# Patterns in the Occurrence and Duration of Musculoskeletal Pain and Interference with Work among Eldercare Workers—A One-Year Longitudinal Study with Measurements Every Four Weeks

**DOI:** 10.3390/ijerph16162990

**Published:** 2019-08-20

**Authors:** Charlotte Diana Nørregaard Rasmussen, Kristina Karstad, Karen Søgaard, Reiner Rugulies, Alex Burdorf, Andreas Holtermann

**Affiliations:** 1National Research Centre for the Working Environment, Lersø Parkallé 105, 2100 Copenhagen Ø, Denmark; 2Department of Sports Science and Clinical Biomechanics and Department of Clinical Research, University of Southern Denmark, Campusvej 55, 5230 Odense M, Denmark; 3Department of Public Health, University of Copenhagen, Nørregade 10, 1165 København, Denmark; 4Department of Psychology, University of Copenhagen, Nørregade 10, 1165 København, Denmark; 5Department of Public Health, Erasmus MC, University Medical Centre, Doctor Molewaterplein 40, 3015 GD Rotterdam, The Netherlands

**Keywords:** prevalence, low back pain, neck pain, shoulder pain, recurrence, recovery, pain episodes

## Abstract

The objective of this study was to examine patterns of musculoskeletal pain episodes over time. We conducted a one-year follow-up study among 275 eldercare workers with measurements of musculoskeletal pain (low back pain (LBP) and neck/shoulder pain (NSP)) and pain-related work interference (PWI) reported via text message every four weeks. We found a constant, high four-weekly prevalence of LBP and NSP (between 61% and 72%). The distributions of pain episodes for LBP and NSP were similar with approximately 30% of the episodes being 7 days or less per four weeks. There was also a high recurrence of pain, with 33% reporting LBP or NSP every four weeks. In addition, 24% had pain at every measurement in both the low back and neck/shoulder regions combined throughout the year. On days with LBP or NSP, approximately 59% also reported interference with work, and 18% of the eldercare workers reported that pain interfered with their work all measurements throughout the year. A high proportion of eldercare workers reported pain every four weeks throughout the year and the four-weekly prevalence of pain remained high and constant on a group level. During most days with pain, eldercare workers were hampered in their regular work activities.

## 1. Introduction

The prevalence of, and burden from, musculoskeletal disorders (MSDs) is high throughout the world with considerable societal costs [1,2,3]. MSDs are estimated to cause 21% of the total years lived with disability [4] and are a main cause of absence from work worldwide [1,3]. Even though low back pain (LBP) and neck/shoulder pain (NSP) are very common health problems, little is known about how they evolve over time. Thus, whether pain is likely to improve, reoccur, persist, or worsen remains an important question. More frequent measurements to track the patterns of pain over time will increase our understanding of MSDs and how they evolve over time [1].

Many previous studies have relied on assessments of MSDs conducted every 3 to 6 months or even yearly [5,6]. Since most pain episodes may be shorter than 3 months, these studies are not likely to capture all fluctuations in MSDs. Text messaging makes it possible to collect data on pain on a monthly, weekly, daily, or even an hourly basis. Text-messaging-based assessment has shown high compliance rates unaffected by age, sex, and season and is also very user friendly for several populations [7,8].

When studying pain, it is important to differentiate pain severity from pain interference. Whereas severity refers to the magnitude of pain, interference refers to pain’s impact on activities [9], including work [10]. While the 12-month prevalence of pain-related interference with activities has been found to be 18% among a general working population [11], few studies have investigated the prevalence of pain-related work interference (PWI) [11]. To our knowledge, no previous studies have investigated PWI with frequent measurements. Thus, there is an urgent need to increase the understanding of MSDs and their consequences for workers, including information about PWI and its patterns over time.

To understand how pain changes over time, we studied LBP, NSP, and PWI over a period of one year through text messages every fourth week among eldercare workers. To our knowledge, this is the first study to investigate the patterns (episodes of pain) of LBP and NSP and PWI in a longitudinal study with a 1-year follow-up period with a high frequency of measurements. More specifically, the aim of this study was to investigate the episodes of LBP, NSP, and PWI in a longitudinal study with a 1-year follow-up period with a high frequency of measurements and an emphasis on prevalence and length of episodes.

## 2. Materials and Methods

The present study was a longitudinal study with a 1-year follow-up period with measurements of MSDs every four weeks among 275 eldercare workers. For this, we used the prospective workplace observational study, the Danish Observational Study of Eldercare work and musculoskeletal disorderS (DOSES), which was designed to examine longitudinal associations between physical and psychosocial working conditions and the occurrence of MSD and its consequences among eldercare workers [12]. The study received ethical approval from the Danish Data Protection Agency and the Ethics Committee for the regional capital of Denmark (H-4-2013-028).

### 2.1. Setting and Participants

The setting for the study was nursing homes in Denmark. We contacted 83 nursing homes by sending emails. Of those, 20 nursing homes responded with interest in the study and were then included in the study. The 20 nursing homes were located in Zeeland in the eastern part of Denmark and had approximately 30–120 residents each. The nursing homes represented both large and small nursing homes as well as both public and private nursing homes. Eligible participants were employed more than 15 h a week, were between 18 to 65 years of age, and provided direct care to residents during at least 25% of their working time. The exclusion criteria to the study were administrative personnel, kitchen and cleaning personnel and nurses, workers employed on night shifts, and workers who were long-term sick-listed, pregnant, or not being permanently employed. Written informed consent was obtained from all participants.

### 2.2. Data Collection Procedure

The baseline data collection ran from September 2013 to December 2014 with nursing homes entering the study at different time points during this time period. At baseline, a structured, self-administered, electronic questionnaire was answered. The questionnaire included socio-demographic measures, i.e., sex, ethnicity (being born in Denmark or not), health and behavior, i.e., questions on pain (days), pain intensity in the previous four weeks on a scale from 0–10 (0 = no pain and 10 = worst pain imaginable, a slightly modified Nordic Musculoskeletal Questionnaire [13]), and smoking. Moreover, body mass index (BMI) (body weight (kg)/(body height (m))^2^) was objectively measured.

A text message survey (SMS) with measurements of pain was conducted for one year at each nursing home starting after the baseline data collection (SMS Track^®^ system(SMS Track APS, Esbjerg, Denmark)) [14]). Every fourth week (on a Monday), the respondents received an automated text message to their private mobile phone that they were expected to answer by using a text message. If an answer was not provided, a reminder was sent on Wednesday. If answers were still missing, we called the participant by phone to get their response.

### 2.3. Outcomes

Every four weeks, LBP was measured by SMS as days with pain in the lower back. The question posed was “During the past four weeks, how many days have you had pain in your lower back region? (Answer from 0–28)”. Likewise, every four weeks, NSP was measured by SMS as days with pain in the neck/shoulder region. The question posed was “During the past four weeks, how many days have you had pain in your neck/shoulder region? (Answer from 0–28)”. Finally, every four weeks, pain-related work interference (PWI) was measured by SMS as days where pain interfered with work. The question posed was “During the past four weeks, how many days due to your pain in your lower back and/or pain in neck/shoulders have you had difficulty performing your work (i.e., affected/complicated the performance of your work tasks)? (Answer from 0–28)” [15]. The question on PWI was only asked if the participants reported pain in either the lower back or neck/shoulder.

### 2.4. Definitions of Episodes of Pain

An episode of pain was defined as occurring at least one day with pain per measurement (four weeks) [2,16,17]. Thus, the four-week prevalence was defined as the proportion of the population experiencing at least one episode of pain at each time point. The one-year prevalence was defined as the proportion of the population experiencing at least one pain episode during the one-year follow-up period. Likewise, a pain-free period consisted of four consecutive weeks without pain [17]. Thus, a period of recovery (recovery) was defined as four weeks (one measurement) without pain following a period with pain and was interpreted as an improvement in pain. Worsening of pain was defined as a measurement with pain following a measurement without pain. Repeated pain/repeated pain free was defined as two or more successive measurements with no change in pain status. Constant pain/constant pain free was defined as all measurements with pain/no pain.

### 2.5. Statistics

We used descriptive techniques to present the episodes of LBP, NSP and PWI over time. To avoid different denominators during the follow-up due to loss during follow-up, the episodes in the presence of LBP, NSP, and PWI were described among participants with a full dataset with all 14 monthly pain measurements throughout the year. We conducted a nonresponse analysis to examine whether those without full answers differed in comparison to those included in the analysis. The episodes of pain were described using the different categories (constant pain free, repeated pain free, improvement, worsening pain, repeated pain, and constant pain). All calculations were performed in IBM SPSS Statistics, version 21 (IBM, Armonk, NY, USA).

## 3. Results

Initially there were 449 workers from 20 different nursing homes included in the text message survey. A full dataset with 14 completed measurements consisted of *n* = 275 for LBP, *n* = 273 for NSP, and *n* = 209 for pain-related work interference (PWI). The discrepancy in the numbers in the datasets with NSP, LBP, and PWI is due to the possibility of having full answers on either LBP, NSP, or PWI and missing answers to the others. Only those with full answers on NSP or LBP were included in the respective analysis. There were no significant differences in age, sex, ethnicity, smoking, body mass index, and musculoskeletal pain between those included in the text message survey (*n* = 449) and those included in the analysis (*n* = 275) (data not shown). The reason for the loss to follow up was mainly due to participants not answering all text messages throughout the year rather than dropping out of the study.

The characteristics of the study population can be seen in Table 1. The study population consisted of mostly females (94%). The average age was 47 years, most participants (83%) were born in Denmark, and 28% reported no LBP or no NSP in the three months before the text message survey. The average LBP intensity (on a scale from 0 to 10) was 4, and the average NSP intensity was 3 (on a scale from 0 to 10) (Table 1).

### 3.1. Duration of Pain (Days per Four Weeks, Days per Year, and Length of Episodes)

The average number of days with LBP per four weeks was 7 (SD: 7.3), and the average number of days with LBP during the one-year follow-up period was 97 (SD: 100.9). Likewise, the average number of days with NSP per four weeks was 7 (SD: 7.5), and the average number of days with NSP during the one-year follow-up was 100 (SD: 105.2). The number of days with PWI per four weeks was 5 (SD: 5.8) and the average number of days with PWI during the one-year follow-up period was 66 (SD: 80.6) (Table 2).

Figure 1 shows the distribution in percentage of length of pain episodes from 1–28 days in all three pain variables (LBP, NSP, and PWI), averaged from 14 four-weekly measurements. No-pain episodes (0 days with pain) were registered in 35% of LBP episodes and in 33% of NSP episodes. The distributions of pain episodes for LBP, NSP, and PWI were similar with approximately 30% of the episodes being 7 days or less. However, approximately 8% of the pain episodes for LBP and NSP were 28 days during the four-week recall period, compared to approximately 4% for PWI.

#### 3.1.1. Patterns of Pain Episodes

Figure 2, Figure 3 and Figure 4 show the distribution of categories of pain over one year. When we combined the three categories of pain (i.e., constant pain, repeated pain, and worsening pain), the four-week prevalence of LBP varied between 61% and 70% with a one-year prevalence of 91%. The four-weekly prevalence of NSP varied between 63% and 72% with a one-year prevalence of 91%. The four-weekly prevalence of reporting combined LBP and NSP varied between 49% and 56% with a one-year prevalence of 79%. Overall, the prevalence was relatively stable over time, but on an individual level, the figures also show great fluctuations in pain status. For instance, workers were relatively stable pain-free over time (repeated pain free) or relatively stable in pain over time (repeated pain). However, many varied between the categories “improvement” and “worsening”. For both LBP and NSP, 33% of the eldercare workers had pain at every measurement throughout the year. In addition, 24% had pain at every measurement in both the low back and neck/shoulder combined throughout the year.

Figure 2 shows the percentages of the population divided into categories of low back pain at each of the 14 measurement points. Improvement is categorized as a measurement with low back pain (an episode of low back pain) followed by a measurement without low back pain (recovery). Worsening is defined as a measurement without low back pain followed by a measurement with low back pain. Repeated pain/repeated pain free in low back is defined as two or more successive measurements with pain/no pain in the lower back.

Figure 3 shows the percentages of the population divided into categories of neck/shoulder pain at each of the 14 measurement points. Improvement is categorized as a measurement with neck/shoulder pain (an episode of neck/shoulder pain) followed by a measurement without neck/shoulder pain (recovery). Worsening is defined as a measurement without neck/shoulder pain followed by a measurement with neck/shoulder pain. Repeated pain/repeated pain free in neck/shoulder is defined as two or more successive measurements with pain/no pain in neck/shoulder.

Figure 4 shows the percentages of the population divided into categories of pain in neck/shoulder and low back at each of the 14 measurement points. Improvement is categorized as a measurement with pain in neck/shoulder and low back (an episode of pain in neck/shoulder and low back) followed by a measurement without pain in neck/shoulder and low back (recovery). Worsening is defined as a measurement without pain in neck/shoulder and low back followed by a measurement with pain in neck/shoulder and low back. Repeated pain/repeated pain free in neck/shoulder and low back is defined as two or more successive measurements with pain/no pain in neck/shoulder and low back.

#### 3.1.2. Pain-Related Work Interference

The distributions of PWI every four weeks were similar for those experiencing LBP and those experiencing NSP. When experiencing LBP or NSP, approximately 59% of participants, on average, experienced interference with work activities (Figure 5). Looking at all eldercare workers, the four-weekly prevalence of PWI varied between 44% and 61% with a one-year prevalence of 88%, and 18% of the eldercare workers reported that pain interfered with their work every four weeks throughout the year (Figure 6).

## 4. Discussion

We found a constant, high four-week prevalence of LBP and NSP. The distribution of pain episodes for LBP and NSP were similar with approximately 30% of the episodes being 7 days or less per four weeks. However, approximately 8% of the pain episodes for LBP and NSP were 4 weeks long. There was also a high recurrence of pain, with 33% of participants reporting LBP or NSP every four weeks and 24% reporting combined LBP and NSP every four weeks. On days with LBP or NSP, approximately 59% of participants also reported interference with work.

### 4.1. Prevalence of LBP and NSP

Our findings suggest that the prevalence of both LBP and NSP in eldercare workers is very high when measured on a monthly basis. The one-year prevalence of LBP was much higher for our population with 91% compared to the annual prevalence which has been found to be 55% among eldercare workers [18] and 38% in the general population, respectively [16]. The four-week prevalence of LBP varied between 61% and 70%, which is much higher than the 31% observed in the general population [16]. The four-week prevalence of NSP varied between 63% and 72% and the one-year prevalence was 91%, which is also higher than the mean annual prevalence in eldercare workers (42%) [18] and in the general working population (27–48%) [19]. Moreover, 33% of the eldercare workers reported LBP and NSP in all measurements throughout the year. In comparison, a study on LBP in the general population found that 12% had pain more or less all of the time [20]. Considering both body regions (NSP and LBP), nearly 60% of the eldercare workers reported pain in at least one region at all measurements. This aligns with a previous study which showed that when present, the pain is persistent rather than episodic [21,22]. The observed yearly prevalence measured with the repeated SMS assessment was high compared to previous findings in the literature. This suggests that either the repeated measures assessments are overestimating or the traditional single retrospective assessments are underestimating the prevalence, or a combination of both possibilities may be occurring. Since the participants were asked to provide baseline information about LBP, NSP, and PWI from the previous 12 weeks, this baseline information was compared with the follow-up 4-weekly measurements to see if the longitudinal results at follow up were higher than those at baseline. Using the same definition for prevalence (< 1 day with pain), the prevalence at baseline for the 12-week period was 72% for both LBP and NSP, which is quite comparable to the monthly prevalence that we found in our study (variation between 61% and 72%).

In this study, we analyzed pain cases every four weeks. Large variations in the case definition of pain exist in the literature [2]. For instance, more than 300 case definitions for neck pain have been identified [23]. Therefore, it is difficult to compare findings because of the use of different questions, variations in case definitions, and other methodological inconsistencies in previous studies. We used the definitions described in the Global Prevalence of Low Back Pain and in the review of the Global burden of neck pain [2,16] yielding findings comparable to those used in large population studies.

### 4.2. Multimorbidity of LBP and NSP

Previous studies have documented that subjects often report musculoskeletal pain in multiple body sites [24,25]. The one-year prevalence of LBP and NSP combined in our study was 79% and the 4-week prevalence of LBP and NSP combined varied between 49% and 56%. This is comparable to the findings in the study by Haukka et al., who found that 83% of their subjects reported pain in both the lower back and neck [24]. In addition, 24% had pain every four weeks in the lower back and neck/shoulder regions combined throughout the year, compared to the 33% reporting either LBP or NSP all measurements throughout the year. These findings imply that pain episodes are not independent.

### 4.3. Pain-Related Work Interference

The one-year prevalence of pain-related work interference (PWI) of 88% is considerably higher than previous findings, where the prevalence of pain-related interference with activities has been found to be 18% in a general working population [11]. In addition, 18% of the eldercare workers reported PWI every four weeks throughout the year. Disability due to pain and duration of pain have been found to be significant predictors for dropout from the eldercare sector 2 years after qualification [26]. We also found that the distributions of PWI every four weeks were similar for those experiencing LBP or NSP. It is surprising that in spite of the experience of interference on the majority of days, this has not been bothersome enough to take sick leave. This is in accordance with the finding in a qualitative study investigating workers’ strategies towards pain. They found that pain made it more difficult to work, but job activities were completed nonetheless [27]. Another study found that among older individuals, health interventions that address chronic musculoskeletal pain may promote subjective health and quality of life as well as improve mental health [28]. To our knowledge, the present study is the first to show the magnitude of pain-related interference with work in eldercare workers and warrants further investigations to find solutions for eldercare workers to be able to remain in their jobs for more years.

### 4.4. Episodes of Pain

In our study, we found some variation in the length of episodes with most episodes being one week or less and only a few being four weeks long. Our study also showed that a substantial proportion of the eldercare workers had some fluctuations with many recurrent episodes of pain during the one-year follow-up period. In comparison, another study among healthcare workers observed that 46% of the subjects with pain varied between being cases or not during a 3-year period [29]. This underlines the need for replacing previous categorizations of pain as being acute, subacute, or chronic in nature with more accurate descriptions of time-varying recurrent and prolonged episodes of pain.

### 4.5. Methodological Issues

Previous studies on the patterns of pain episodes have mostly relied only on a few measurements taken over time interspersed by long intervals, e.g., years [6]. Such studies may not capture detailed patterns or fluctuations in pain. In contrast, frequent, repeated measurements of pain facilitate accurate and precise identification of individual patterns of pain episodes [30] and minimize recall bias [31,32]. The use of measurements of pain every four weeks in the current study is therefore a major strength.

A limitation in this study is the combined NSP question, which did not separate the neck region from the shoulder region. However, a previous study showed a high concurrence of neck and shoulder complaints [6]. We only included between 47% and 61% of the original study population in our analysis, which may have given rise to selection bias. However, baseline measurements of pain were almost exactly similar to those during repeated follow-up, and we did an analysis to check for differences in demographic and pain variables between the two populations and found no statistical differences between them, suggesting that the reported pain was not substantially influenced by selective participation. Another limitation is the lack of a comparison group to compare the prevalence of pain in working populations without physically strenuous working conditions.

We followed the eldercare workers every four weeks throughout one year to establish the patterns of musculoskeletal pain episodes. We chose to use simple descriptive statistics to describe the patterns of pain episodes in detail. We found that pain was either persistent or episodic rather than presenting in just one well-defined episode, and therefore, single time-point outcomes are not optimal measures of musculoskeletal pain. Cohort studies on risk factors for the prevalence of pain should not only pay attention to new episodes but should also analyze risk factors for constant pain, constant pain-free status, as well as fluctuations in pain (e.g., improvement and worsening). Also, the use of musculoskeletal pain as an outcome in intervention studies should be revisited, including considering an increase in the follow-up period to ensure well-described patterns of pain episodes before and after an intervention are collected.

## 5. Conclusions

In summary, LBP and NSP among eldercare workers were characterized by high four-week and yearly prevalence. Although, at the population level, the prevalence of pain remained constant, the patterns of pain episodes within individuals demonstrated strong fluctuations. Finally, when studying pain, it is important to differentiate between pain and functional limitations due to pain. In our study, we investigated pain-related interference with work. An important finding of our study is that, on most days with pain, there was also interference with work. More studies on pain-related work interference (PWI) are needed to establish prognostic factors and to improve diagnosis and treatment when pain is so severe that it impacts an individual’s ability to work. We also need more studies to evaluate whether PWI is more predictive of future sick leave and work ability than LBP and NSP alone.

## Figures and Tables

**Figure 1 ijerph-16-02990-f001:**
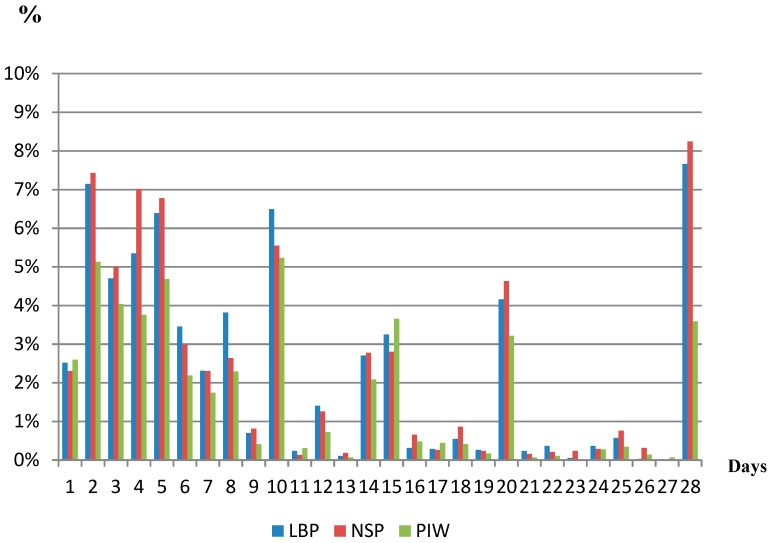
Distribution in percentage of the study population of the length of pain episodes lasting from 1 to 28 days. LBP: low back pain; NSP: neck/shoulder pain; PWI: pain-related interference with work.

**Figure 2 ijerph-16-02990-f002:**
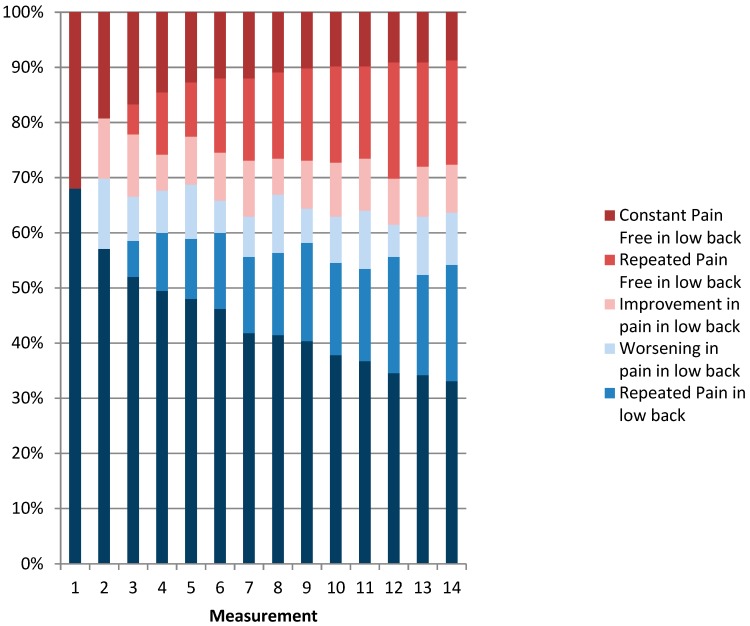
Patterns of low back pain episodes over one year.

**Figure 3 ijerph-16-02990-f003:**
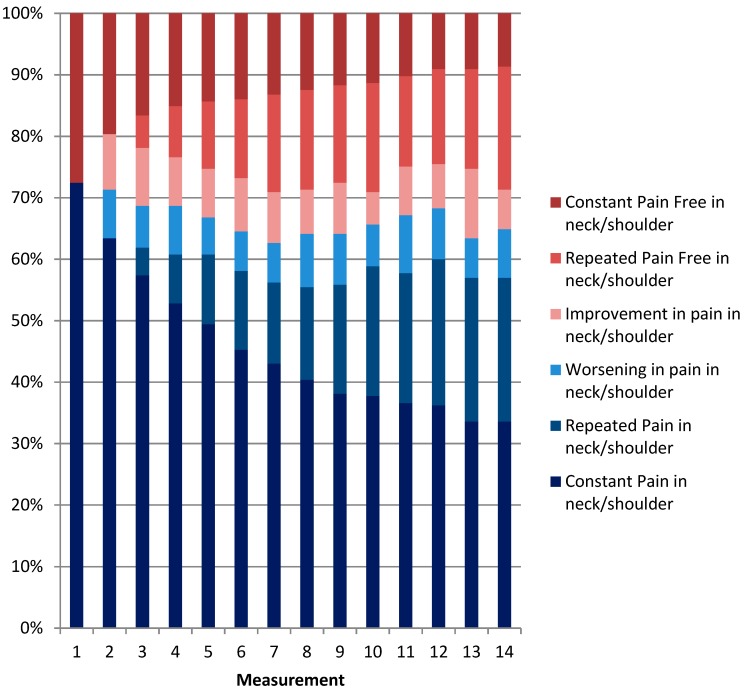
Patterns of neck/shoulder pain episodes over one year.

**Figure 4 ijerph-16-02990-f004:**
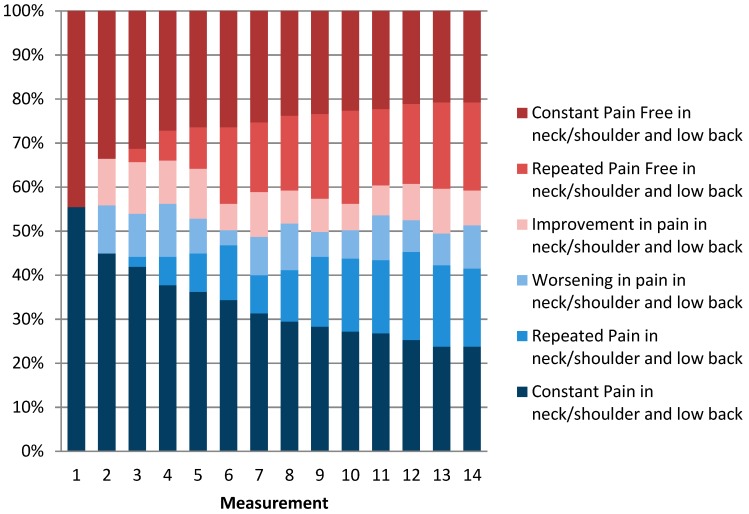
Patterns of combined low back pain and neck/shoulder pain episodes over one year.

**Figure 5 ijerph-16-02990-f005:**
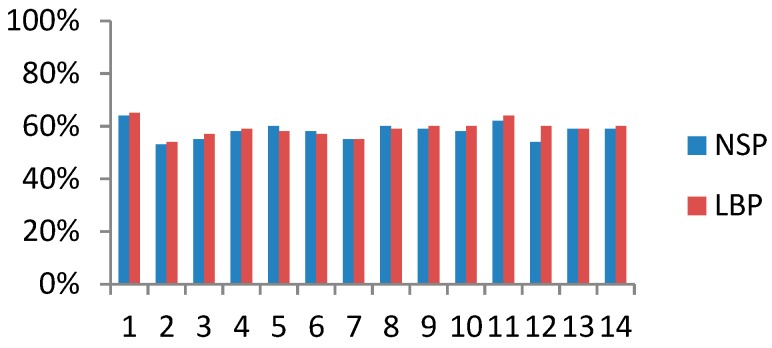
Distribution of pain-related work interference among those experiencing NSP and LBP.

**Figure 6 ijerph-16-02990-f006:**
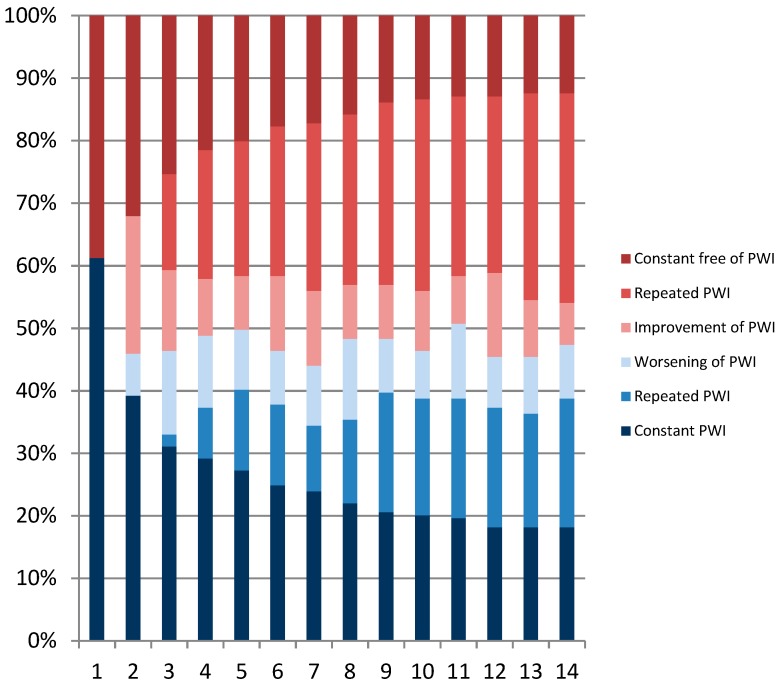
Patterns of episodes of pain-related work interference over one year. The figure shows the percentages of the population divided into different categories at each of the 14 measurement points. Improvement is categorized as a measurement with pain-related work interference (PWI) (an episode with PWI) followed by a measurement without PWI (recovery). Worsening is defined as a measurement without PWI followed by a measurement with PWI. Repeated PWI/repeated PWI free is defined as two or more successive measurements with PWI/no PWI.

**Table 1 ijerph-16-02990-t001:** Baseline characteristics of 275 eldercare workers in 20 nursing homes in Denmark.

Baseline Characteristic	Mean	Numbers
	(SD)	(%)
**Age (years)**	47 (10.4)	
**Sex (female)**		259 (94%)
**Ethnicity (born in Denmark)**		228 (83%)
**Smokers**		95 (36%)
**Body mass index (kg/m^2^)**	27 (5.4)	
**Low back pain previous 12 weeks**		
0 days		74 (28%)
1–2 days		28 (11%)
3–7 days		53 (20%)
8–14 days		37 (14%)
15–30 days		28 (11%)
31–60 days		21 (8%)
61–84 days		25 (9%)
**Neck/shoulder pain previous 12 weeks**		
0 days		74 (28%)
1–2 days		32 (12%)
3–7 days		52 (20%)
8–14 days		26 (10%)
15–30 days		36 (14%)
31–60 days		20 (8%)
61–84 days		27 (10%)
**Low back pain intensity (0–10)**	4 (2.3)	
**Neck/shoulder pain intensity (0–10)**	3 (2.4)	

Data are mean (SD) or numbers (%).

**Table 2 ijerph-16-02990-t002:** Average duration of low back pain and neck/shoulder pain and pain-related interference with work (total days per four weeks and total days per year).

Presence of Musculoskeletal Complaints per Person	Mean	Median (Interquartile Range)
Low back pain (total days per year)	97	98 (128)
Low back pain (days per four weeks)	7	7 (9.14)
Neck/shoulder pain (total days per year)	100	83 (134)
Neck/shoulder pain (days per four weeks)	7	6 (10.04)
Pain-related interference with work (total days per year)	66	30 (95.00)
Pain-related interference with work (days per four weeks)	5	2 (6.79)

Data are means and medians (interquartile range).

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
