# Peer review of "Patterns in the Occurrence and Duration of Musculoskeletal Pain and Interference with Work among Eldercare Workers—A One-Year Longitudinal Study with Measurements Every Four Weeks"

_ijerph, 2019, doi:10.3390/ijerph16162990_

Round 1

Reviewer 1 Report

This paper is a descriptive prospective study of MS pain in eldercare workers. The study uses a novel method to measure the prevalence of pain among workers, namely using short message service (SMS) to ask about pain every four weeks. Although the paper is well-written (with only minor typographical errors), due to the study design, I do not see what this paper adds to the current body of knowledge.

1) The explorative study and its methods do not help explain the occupational causes of pain or help indicate possibilities for prevention.

2) Without a comparison group, it is difficult to say if this method of assessing pain (per SMS) would not also fund comparably increased a prevalence of pain in other working populations without physically strenuous working conditions, such as administrative office workers.

3) The observed yearly prevalence measured with the repeated SMS assessment was extremely high compared to previous research findings. This suggests that this method may be overestimating or traditional single retrospective assessments may be underestimating the prevalence, or both might be occurring.

    a) It seems the SMS assessment did not use any threshold of pain to determine the number of days in the past 28 days where pain was experienced. Could this have resulted in the over reporting (i.e. muscle tension reported as pain). Also could the assessment method itself be causing more pain by directing attention to pain? Pain has a substation psychological component, and repeatedly asking workers to think about how often they experience pain could be lowering workers' pain threshold and resulting in this increased prevalence. Did reported days of pain and periods increase over time during the observation period?

    b) But most previous studies use one retrospective assessment, and this form of assessment might be impacted by recall bias, where workers may not recall minor pain episodes after a few months, especially if the pain did not result in any medical leave or a doctor's visit. 

Unfortunately, while the study results imply that these two things may be occurring, the methods used here do not provide any evidence to elucidate what is happening. Perhaps using a control group from the same population asked with a typical retrospective assessment could be useful here.

4) Although the authors claim have assessed if the persons lost to follow up were similar to the original population, very few of the original population were included in the final analysis. It would be important to help the readers judge if some of this loss is due to healthy worker bias. This could be made possible with reporting how many of respondents were excluded because of going on sick-leave, changing jobs, or just not reporting back on all 14 assessments. Also, if this method of assessment is to be used in future studies, some method of using is required to still be able to include persons that miss one assessment (i.e. imputation or asking about the last 8 weeks at the next assessment).

5) It might also be good further test the assessment validity to see if different answers are provided on different days of the week. 

6) How many workers agreed to take part in the study, i.e. what was the initial response (%)?

Author Response

This paper is a descriptive prospective study of MS pain in eldercare workers. The study uses a novel method to measure the prevalence of pain among workers, namely using short message service (SMS) to ask about pain every four weeks. Although the paper is well-written (with only minor typographical errors), due to the study design, I do not see what this paper adds to the current body of knowledge.

Thank you, for your comments. We agree that this is a novel approach to pain measurements. We believe that this adds new knowledge about the magnitude of pain for this particular job group. In addition this study contributes with important knowledge about how pain develops over time, affects work and challenges the usual definitions of pain episodes.

The explorative study and its methods do not help explain the occupational causes of pain or help indicate possibilities for prevention.

We agree, but it was outside the scope of this paper. However, the knowledge from this study can contribute to identifying groups and indicate new possibilities for prevention.

Without a comparison group, it is difficult to say if this method of assessing pain (per SMS) would not also fund comparably increased a prevalence of pain in other working populations without physically strenuous working conditions, such as administrative office workers.

We agree that this is an interesting research question which requires data on other job groups with less strenuous working conditions as well. Unfortunately, this was not possible based on the existing dataset. We have added this as a limitation in our study.

The observed yearly prevalence measured with the repeated SMS assessment was extremely high compared to previous research findings. This suggests that this method may be overestimating or traditional single retrospective assessments may be underestimating the prevalence, or both might be occurring.

Yes, that is true. However, using the same definition for prevalence (>1 day with pain), then the prevalence at baseline for the 12 weeks were 72% for both LBP and NSP, which is quite comparable to the monthly prevalence that we find in our study (varies between 61-72%). We have elaborated on this in the discussion.

a) It seems the SMS assessment did not use any threshold of pain to determine the number of days in the past 28 days where pain was experienced. Could this have resulted in the over reporting (i.e. muscle tension reported as pain). Also could the assessment method itself be causing more pain by directing attention to pain? Pain has a substation psychological component, and repeatedly asking workers to think about how often they experience pain could be lowering workers' pain threshold and resulting in this increased prevalence. Did reported days of pain and periods increase over time during the observation period?

We used the current definition of how to define an episode of pain by using days with pain, which has been used for all the burden of disease studies reporting prevalence of pain. It has also been advised that pain measurements should be taken frequently, every 4 weeks, as we have done in this study or even more frequently. The prevalence is rather stable over time (only varying between 61-72%) in our study, which supports that the assessment method did not influence the reported days and periods of pain over time.

  b) But most previous studies use one retrospective assessment, and this form of assessment might be impacted by recall bias, where workers may not recall minor pain episodes after a few months, especially if the pain did not result in any medical leave or a doctor's visit. 

Yes, it has previously been shown that use of retrospective measurements might be impacted by recall bias. This is one of the reasons why repeated measurements with short recall period of pain are recommended.

Unfortunately, while the study results imply that these two things may be occurring, the methods used here do not provide any evidence to elucidate what is happening. Perhaps using a control group from the same population asked with a typical retrospective assessment could be useful here.

We agree that a control group could have been beneficial, but we do not see this as necessary for answering the main research question of our study, being to examine pain patterns over time in this specific occupational group.

Although the authors claim have assessed if the persons lost to follow up were similar to the original population, very few of the original population were included in the final analysis. It would be important to help the readers judge if some of this loss is due to healthy worker bias. This could be made possible with reporting how many of respondents were excluded because of going on sick-leave, changing jobs, or just not reporting back on all 14 assessments. Also, if this method of assessment is to be used in future studies, some method of using is required to still be able to include persons that miss one assessment (i.e. imputation or asking about the last 8 weeks at the next assessment).

We see the reviewers concern. However, the main reason for the lost to follow up is due to missing one or more answers of the 14 text messages throughout the year rather than dropping out of the study for one or the other reason. We have now more clearly explained this in the manuscript.

It might also be good further test the assessment validity to see if different answers are provided on different days of the week. 

That is also a good point that should be investigated in future studies. Unfortunately, the research design and data in the present study does not allow such an analysis.

How many workers agreed to take part in the study, i.e. what was the initial response (%)?

Initially, 449 workers were included in the text message survey, but only those with full answers on all 14 pain reportings throughout the year were included in the study. This information has now been added to the manuscript.

Reviewer 2 Report

Overall comments:

This was a longitudinal study with a one-year follow-up with participants recording their MSD every four weeks among 275 eldercare workers. Most of the studies in the area used 7-day, 30-day and 12-month prevalence to measure workers' MSD, which would have recall bias. The results of this study would minimize the recall bias by inviting the participants to report their MSD every four week through SMS. Although the MSD issues are well known in health care workers, particularly nursing assistants, the results would add knowledge on the prevalence of the problems.

Specific comments:

(1) Material and methods: 

(a) 2.1. - p.2, line 67 - please clarify the number of nursing homes involved. Would it be 275 eldercare workers from 20 nursing homes?

(b) 2.2 - p.2, line 77 - please explain what "stepwise entry" means.

(c) 2.5 -p.3, line 116 - please clarify what "14 completed measurements" mean. On page 4, line 141, "14 four-weekly measurements" was used. One year has 52 weeks + 2 days, so it would be 13 weeks + 2 days. Do you mean the 14 completed measurements would be 13 weeks + 2 days measurements? Since only 2 days in 14th week, should 13 completed measurements be used?

(2) Results

(a) p.3, line 125 - please clarify what is the difference between "text message survey" and "those included in the analysis". Should they be the same group of participants?

(3) Discussion

Comparison of the study results with previous studies has been performed. Since the participants were asked to provide baseline information of LBP, NSP and PWI on previous 12 weeks, please compare this baseline information with the follow-up 4-weekly measurements to see if the longitudinal results were higher than the baseline. 

Author Response

This was a longitudinal study with a one-year follow-up with participants recording their MSD every four weeks among 275 eldercare workers. Most of the studies in the area used 7-day, 30-day and 12-month prevalence to measure workers' MSD, which would have recall bias. The results of this study would minimize the recall bias by inviting the participants to report their MSD every four week through SMS. Although the MSD issues are well known in health care workers, particularly nursing assistants, the results would add knowledge on the prevalence of the problems.

Thank you for the summary and the following comments.

Specific comments:

(1) Material and methods: 

(a) 2.1. - p.2, line 67 - please clarify the number of nursing homes involved. Would it be 275 eldercare workers from 20 nursing homes?

Yes, the 275 eldercare workers were from 20 different nursing homes. This has now been clarified in the manuscript.

(b) 2.2 - p.2, line 77 - please explain what "stepwise entry" means.

This simply means that the nursing homes entered the study at different timepoints from September 2013 to December 2014. This has now been clarified in the manuscript.

(c) 2.5 -p.3, line 116 - please clarify what "14 completed measurements" mean.

It means that the participants should have answered all 14 monthly pain measurements throughout the year to be included in the analysis. This has now been clarified in the manuscript.

On page 4, line 141, "14 four-weekly measurements" was used. One year has 52 weeks + 2 days, so it would be 13 weeks + 2 days. Do you mean the 14 completed measurements would be 13 weeks + 2 days measurements? Since only 2 days in 14th week, should 13 completed measurements be used?

We used 14 measurements: we measured every 4 weeks, so to be clear the measurements took a little more than a year.

(2) Results

(a) p.3, line 125 - please clarify what is the difference between "text message survey" and "those included in the analysis". Should they be the same group of participants?

The 275 included in the analysis is a subpart of the larger sample invited to take part in the text message survey (n=449). We compared these two populations to see whether they differed from each other, which was not the case. This has now been clarified.

(3) Discussion

Comparison of the study results with previous studies has been performed. Since the participants were asked to provide baseline information of LBP, NSP and PWI on previous 12 weeks, please compare this baseline information with the follow-up 4-weekly measurements to see if the longitudinal results were higher than the baseline. 

Using the same definition for prevalence (<1 day with pain), then the prevalence at baseline for the 12 weeks, were 72% for both LBP and NSP, which is quite comparable to the monthly prevalence (varies between 61-72%)  that we find over the time course of our study.  We have added this information to the manuscript in the discussion.

Reviewer 3 Report

Overall this paper represents a nice effort, and there is a lot of information for the reader to digest. A couple of things stand out right away, the first is that I had to go back and look up PWI to find it defined. If the abbreviation is to be used it should perhaps be re-defined in each section.

I'm unclear as to how the nursing homes were selected and out of what pool. How many nursing homes are there in the Zeeland area from which the 20 were selected? How were they selected?

There is no notation of human subject/ ethical review for the study protocol. If the authors feel it is exempt they should say so, otherwise they should describe the oversight.

The exceedingly high rate of symptoms in the cohort requires some explanation or further thought in the manuscript - do we think this is this due to injury? Is lifting common in this population?

Why is there a discrepancy in the number of NSP and LBP full data sets?

This statement in the introduction:

"Even though low back pain (LBP) and neck/shoulder pain (NSP) are very common health problems, little is known about how they develop over time. Thus, whether pain is likely to improve, reoccur, persist, or worsen, remains an important question." 

Is unsupported by citation, and unclear. Chronic neck problems develop commonly after neck trauma, as does LBP (the way this is written it is confusing as to whether the authors are stating that we don't know what causes neck and back problems [incorrect] or how such conditions evolve over time [more correct]). Please clarify.

The following sentence:

 More frequent measurements to track the patterns of pain over time will increase our understanding of MSDs 

Doesn't clearly follow anything in the prior paragraph (more frequent than what? Understanding of what parameter of MSDs?)

Author Response

Overall this paper represents a nice effort, and there is a lot of information for the reader to digest. A couple of things stand out right away, the first is that I had to go back and look up PWI to find it defined. If the abbreviation is to be used it should perhaps be re-defined in each section.

We have now defined the abbreviations in each of the sections.

I'm unclear as to how the nursing homes were selected and out of what pool. How many nursing homes are there in the Zeeland area from which the 20 were selected? How were they selected?

We contacted 83 nursing homes by sending an email. Of those 20 nursing homes responded with interest in the study and were then included. This information is now added to the manuscript.

There is no notation of human subject/ ethical review for the study protocol. If the authors feel it is exempt they should say so, otherwise they should describe the oversight.

The study received ethical approval from the Danish Data Protection Agency and the Ethics Committee for the regional capital of Denmark (H-4-2013-028). We apologize for not having provided this and have now added this information to the manuscript.

The exceedingly high rate of symptoms in the cohort requires some explanation or further thought in the manuscript - do we think this is this due to injury? Is lifting common in this population?

We don’t know the cause to the high prevalence of symptoms based on our study, but it is not likely to be caused by a single factor like lifting, but multiple biopsychosocial factors at work and probably outside work. This is something that will be investigated in future studies using data from the DOSES cohort.

Why is there a discrepancy in the number of NSP and LBP full data sets?

This is due to the possibility of full answers on either LBP or NSP and missing answers to the other. Only those with full answers on NSP or LBP were included in the respective analysis. We have clarified this in the manuscript.

This statement in the introduction:

"Even though low back pain (LBP) and neck/shoulder pain (NSP) are very common health problems, little is known about how they develop over time. Thus, whether pain is likely to improve, reoccur, persist, or worsen, remains an important question." 

Is unsupported by citation, and unclear. Chronic neck problems develop commonly after neck trauma, as does LBP (the way this is written it is confusing as to whether the authors are stating that we don't know what causes neck and back problems [incorrect] or how such conditions evolve over time [more correct]). Please clarify.

We agree, and it is how conditions evolve over time that we mean. This has now been clarified.

The following sentence:

 More frequent measurements to track the patterns of pain over time will increase our understanding of MSDs 

Doesn't clearly follow anything in the prior paragraph (more frequent than what? Understanding of what parameter of MSDs?)

This has now been clarified: “Even though low back pain (LBP) and neck/shoulder pain (NSP) are very common health problems, little is known about how they evolve over time. Thus, whether pain is likely to improve, reoccur, persist, or worsen, remains an important question. More frequent measurements to track the patterns of pain over time will increase our understanding of MSDs and how it evolves over time [1].”

Reviewer 4 Report

General Comments

This study investigated the patterns of low back pain and neck/shoulder pain and pain-related work interference in a longitudinal study with a 1-year 56 follow-up with a high frequency of measurements. It explored the new knowledge for the study community who focused on the pain research.  However, the suggestions in details need to be considered as followed.

Details Comments

Introduction:

Please add a sentence in line 51 to explain the difference between this study and the previous researches on PWI, in order to highlight the innovations.

Methods:

Please explain how to measure the intensity of low back pain and neck/shoulder pain

Results

In the beginning of 3.1. Duration of pain, please define what is the pattern of pain, what it contains, what the characteristics are, and how to react and measure it. It is recommended to merge this definition with2.4Definitions of episodes of pain.

Please illustrate add Figure 2b (Line 171) and Figure 2c (Line 177).

Discussion:

The article aims to examine the vertical correlation between physical and psychosocial working conditions, as well as the occurrence of MSD and its impact on care workers. This is the main issue of the article: the correlationand impact should be more focused, while the current discussion is more about current state of pain/morbidity/onset.

Please cite the article “Chronic Musculoskeletal Pain, Self-Reported Health and Quality of Life among Older Populations in South Africa and Uganda” in line 249. It revealed “Among older individuals, health interventions that address CMP may help promote subjective health and quality and life and improve psychological health”

Author Response

This study investigated the patterns of low back pain and neck/shoulder pain and pain-related work interference in a longitudinal study with a 1-year 56 follow-up with a high frequency of measurements. It explored the new knowledge for the study community who focused on the pain research.  However, the suggestions in details need to be considered as followed.

We thank you for your comments and have revised the paper accordingly.

Details Comments

Introduction:

Please add a sentence in line 51 to explain the difference between this study and the previous researches on PWI, in order to highlight the innovations.

This has now been clarified: While the 12-month prevalence of pain-related interference with activities has been found to be 18% among a general working population [11], only few studies have investigated the prevalence of pain-related work interference (PWI) [11]. To our knowledge no previous studies have investigated PWI with frequent measurements. Thus, there is an urgent need to increase the understanding of MSD’s and its consequences for workers, including information of PWI and its patterns over time.

Methods:

Please explain how to measure the intensity of low back pain and neck/shoulder pain

This is explained on page 2 line 80-82 : pain intensity previous four weeks on a scale from 0-10 (0= no pain and 10=worst pain imaginable)(a slightly modified Nordic Musculoskeletal Questionnaire [13]

Results

In the beginning of 3.1. Duration of pain, please define what is the pattern of pain, what it contains, what the characteristics are, and how to react and measure it. It is recommended to merge this definition with2.4Definitions of episodes of pain.

Thanks for pointing this out. We can see that there is some inconsistency in our use of “patterns of pain” and “episodes of pain” and we have changed this throughout the manuscript.

Please illustrate add Figure 2b (Line 171) and Figure 2c (Line 177).

This has now been added.

Discussion:

The article aims to examine the vertical correlation between physical and psychosocial working conditions, as well as the occurrence of MSD and its impact on care workers. This is the main issue of the article: the correlation and impact should be more focused, while the current discussion is more about current state of pain/morbidity/onset.

The current study was designed to investigate the patterns of musculoskeletal pain episodes over time. In accordance the discussion, is mainly focused on this.

The cohort providing the data was designed to investigate longitudinal associations between physical and psychosocial working conditions and occurrence of MSD and its consequences among eldercare workers [12] and over time will investigate these issues.

Please cite the article “Chronic Musculoskeletal Pain, Self-Reported Health and Quality of Life among Older Populations in South Africa and Uganda” in line 249. It revealed “Among older individuals, health interventions that address CMP may help promote subjective health and quality and life and improve psychological health”

Thank you for informing us about this paper. We have now added it as a reference.

Round 2

Reviewer 1 Report

Thank you for your revisions. I do think the revisions improve the paper. As a descriptive/explorative paper of musculoskeletal pain (MSP) prevalence in eldercare workers, it is acceptable. There is one further issue that should be considered:

The conclusions stated in the lines 243-246 beginning with “Our findings suggest that the prevalence of both LBP and NSP is higher than previously estimated,” seem too bold and should be reconsidered. For one, the review cited by Davis and Kotowski 2015 [18] to come to this conclusion report an average 12-month prevalence of 55% for LBP among nurses in general, not just eldercare workers. Then it is unclear how exactly Davis and Kotowski derived this average, and if the prevalence of each individual study included in the review’s average was weighted according to the size of the population. This makes the 55% 12-month prevalence estimate in [18] questionable. From the figure 1 in the Davis and Kotowski paper, the 12-month prevalence of LBP ranged from around 10-90% (!), which seems be in accordance with your findings. Also Davis and Kotowski averaged international results (some from over 20 years ago) without considering any sources of heterogeneity. Is it possible that working populations in Europe may be older on average and working conditions may differ compared to the conditions on other continents included in the 55% estimate? Are there are any more comparable population estimates from Europe or Denmark that could also be cited, which could provide a more valid comparison for this population of Danish eldercare workers? Other than various methods of measuring MSD, are there are reasons that could explain the increased 12-month prevalence of LBP you found?

Davis KG, Kotowski SE: Prevalence of Musculoskeletal Disorders for Nurses in Hospitals, Long-Term Care Facilities, and Home Health Care A Comprehensive Review. Human Factors: The Journal of the Human Factors and Ergonomics Society 2015:0018720815581933

Author Response

Thank you for your revisions. I do think the revisions improve the paper. As a descriptive/explorative paper of musculoskeletal pain (MSP) prevalence in eldercare workers, it is acceptable. There is one further issue that should be considered:

The conclusions stated in the lines 243-246 beginning with “Our findings suggest that the prevalence of both LBP and NSP is higher than previously estimated,” seem too bold and should be reconsidered. For one, the review cited by Davis and Kotowski 2015 [18] to come to this conclusion report an average 12-month prevalence of 55% for LBP among nurses in general, not just eldercare workers. Then it is unclear how exactly Davis and Kotowski derived this average, and if the prevalence of each individual study included in the review’s average was weighted according to the size of the population. This makes the 55% 12-month prevalence estimate in [18] questionable. From the figure 1 in the Davis and Kotowski paper, the 12-month prevalence of LBP ranged from around 10-90% (!), which seems be in accordance with your findings. Also Davis and Kotowski averaged international results (some from over 20 years ago) without considering any sources of heterogeneity. Is it possible that working populations in Europe may be older on average and working conditions may differ compared to the conditions on other continents included in the 55% estimate? Are there are any more comparable population estimates from Europe or Denmark that could also be cited, which could provide a more valid comparison for this population of Danish eldercare workers? Other than various methods of measuring MSD, are there are reasons that could explain the increased 12-month prevalence of LBP you found?

We thank the reviewer and appreciate that our previous comments and adjustments has been satisfactory. 

We understand your point. However, since many of the previous studies have not used either the same definition of pain (e.g. many have used intensity of pain) or have not used frequent measures, it is very difficult to compare the prevalence in these studies to ours. However, we agree about the conclusion and have now changed the sentence in this paragraph: "Our findings suggest that the prevalence in eldercare workers of both LBP and NSP when measured on a monthly basis is very high".